# Tumor-targeted silencing of the peptide transporter TAP induces potent antitumor immunity

Greta Garrido[1], Brett Schrand[1], Ailem Rabasa[1], Agata Levay[1], Francesca D'Eramo[1], Alexey Berezhnoy[1], Shrey Modi[2], Tal Gefen[1], Koen Marijt [iD] [3], Elien Doorduijn[3], Vikas Dudeja[2], Thorbald van Hall [iD] [3] & Eli Gilboa[1]

Neoantigen burden is a major determinant of tumor immunogenicity, underscored by recent clinical experience with checkpoint blockade therapy. Yet the majority of patients do not express, or express too few, neoantigens, and hence are less responsive to immune therapy. Here we describe an approach whereby a common set of new antigens are induced in tumor cells in situ by transient downregulation of the transporter associated with antigen processing (TAP). Administration of TAP siRNA conjugated to a broad-range tumor-targeting nucleolin aptamer inhibited tumor growth in multiple tumor models without measurable toxicity, was comparatively effective to vaccination against prototypic mutation-generated neoantigens, potentiated the antitumor effect of PD-1 antibody or Flt3 ligand, and induced the presentation of a TAP-independent peptide in human tumor cells. Treatment with the chemically-synthesized nucleolin aptamer-TAP siRNA conjugate represents a broadly-applicable approach to increase the antigenicity of tumor lesions and thereby enhance the effectiveness of immune potentiating therapies.

---

[1] Department of Microbiology and Immunology, University of Miami, Miller School of Medicine, Miami, FL, USA. [2] Department of Surgery, University of Miami, Miller School of Medicine, Miami, FL, USA. [3] Department of Medical Oncology, Leiden University Medical Center, Leiden, Netherlands. Correspondence and requests for materials should be addressed to E.G. (email: egilboa@med.miami.edu)

Responsiveness to checkpoint blockade therapy was shown to correlate with the number of neoantigens expressed in the patient' tumors[1,2]. Yet, many tumors do not express or express too few neoantigens[3,4], and therefore the majority of patients will be less responsive to checkpoint blockade or other forms of immune therapy[5,6]. Here we describe an approach to induce the expression of new antigens in the patients' disseminated tumor lesions that will be applicable also to the majority of the patients that do not express tumor resident mutation-generated neoantigens.

We have previously described an approach to induce neoantigens in tumor cells by inhibiting the nonsense-mediated mRNA decay (NMD) pathway[7], but it's clinical applicability was brought into question because the majority of the induced neoantigens will be tumor cell-specific[8]. An alternative way of inducing new antigens was suggested by the studies of Van Hall and colleagues showing that genetic ablation of transporter associated with antigen processing (TAP) not only downregulates the canonical MHC class I presentation pathway, but also leads to the presentation of new class I-restricted epitopes encoded in housekeeping products. The resulting MHC–peptide complexes were recognized by CD8+ T cells and inhibited the growth of TAP deficient, but not TAP sufficient, tumors[9,10]. Importantly, since the TAP deficiency-induced antigens are not generated as a result of random mutations in the cell they are more likely to be shared among all (tumor) cells in which TAP was downregulated, as was indeed shown[11–13], corresponding to spontaneously arising, albeit rare, clonal tumor-expressed neoantigens that are more likely to elicit a protective immune response[5,14–16].

With the aim of developing a clinically translatable and broadly applicable approach to induce TAP deficiency-induced neoantigens in tumor cells in vivo, we used siRNA to downregulate TAP that was targeted to murine and human tumor cells by conjugation to a broad-range nucleolin-binding oligonucleotide aptamer. The nucleolin aptamer used in our study as prototype of tumor-targeting ligand, exhibits specificity to a broad range of tumor cells of both murine and human origin, and was used to target biological agents to tumor cells in mice[17].

## Results

**Aptamer-targeted siRNA downregulation of TAP in tumor cells.** We first confirmed that the AS1411 nucleolin aptamer used in our study[17] conjugated to a TAP siRNA (Supplementary Fig. 1a) accumulated preferentially in the subcutaneously growing 4T1 breast carcinoma tumors (Supplementary Fig. 1b). Incubation of RMA T lymphoma ($H-2^b$), A20 B lymphoma ($H-2^d$), or 4T1 ($H-2^d$) cells in vitro with nucleolin aptamer conjugated to a TAP siRNA (Nucl-TAP), but not with nucleolin aptamer conjugated to a nonfunctional siRNA (Nucl-Ctrl) or with a nonbinding aptamer conjugated to TAP siRNA (Ctrl-TAP), led to downregulation of TAP mRNA (Fig. 1a) in the absence of measurable cytotoxicity (Supplementary Fig. 2). In contrast, Nucl-TAP did not reduce TAP expression in nontransformed NIH 3T3 cells (Fig. 1a). As first described in TAP-deficient cells[18], nucleolin aptamer-targeted siRNA inhibition of TAP led to the downregulation of MHC class I $D^d$ and $K^b$, but not $Qa-1^b$, alleles (Fig. 1b). Furthermore, activation of a previously described T cell clone recognizing the TAP-deficient epitope TRH4[13] was comparable in RMA cells treated with Nucl-TAP to RMA cells pulsed with TRH4 peptide, or the TAP-deficient RMA-S cell line (Fig. 1c). As shown in Fig. 1d, the TRH4 peptide was also presented by MC38 and GL261 tumor cells incubated with Nucl-TAP, consistent with the hypothesis that new epitopes presented upon TAP downregulation could function as shared, non-mutated clonal neoantigens.

We next tested the efficiency of Nucl-TAP delivery to GFP-expressing A20 lymphoma-bearing mice by intraperitoneal administration of the aptamer-siRNA conjugates. As shown in Fig. 2a, treatment with Nucl-TAP, but not Nucl-Ctrl, led to a 35–45% reduction of TAP mRNA in the sorted population of GFP-expressing cells that persisted for 3–4 days. Measuring TAP downregulation-induced reduction of MHC class Ia expression by flow cytometry shows that Nucl-TAP treatment led to partial reduction of MHC class Ia expression in the majority of tumor cells that also persisted for 3–4 days following Nucl-TAP treatment (Fig. 2b, c). MHC class Ia was downregulated in the GFP-positive tumor cells but not in the GFP-negative tumor-resident endothelial cells, leukocytes, or fibroblasts (Fig. 2d).

Taken together, these experiments show that Nucl-TAP downregulates TAP in tumor cells in vitro and in vivo, and recapitulates the phenotype of TAP-deficient cells in terms of selective downregulation of MHC class Ia expression and presentation of a common novel epitope.

**TAP downregulation in tumor cells inhibits tumor growth in mice.** Consistent with the hypothesis that Nucl-TAP transiently inhibits TAP mRNA and induces the expression of new antigens in tumor lesions, treatment of subcutaneously implanted palpable 4T1 tumor-bearing Balb/c mice with Nucl-TAP, but not with Nucl-Ctrl or Ctrl-TAP, inhibited tumor growth (Fig. 3a), and could be further enhanced by co-treatment with PD-1 Ab or Flt3 ligand (Fig. 3b), suggesting that increasing the antigenic burden of tumor cells by transient TAP downregulation could enhance the effectiveness of immune potentiating therapies, not limited to checkpoint blockade. Transient tumor-targeted TAP downregulation also inhibited the growth of palpable RMA (Supplementary Fig. 3a), as well as A20 tumors in which case a proportion of mice rejected the tumors completely (11/30), reflecting the enhanced immunogenicity of A20 compared to 4T1 tumor (Fig. 3c). The generality of inducing new antigens by targeting key mediators of antigen processing is suggested by the observation that downregulation of the ER aminopeptidase associated with antigen processing (ERAAP) also inhibits tumor growth and synergizes with PD-1 Ab treatment (Supplementary Fig. 3b). Nucl-TAP treatment inhibited local tumor growth (Fig. 3d) and reduced metastasis (Supplementary Fig. 3d, e) in an orthotopic model for pancreatic cancer[19] that was comparable to treatment with Minnelide (Fig. 3e and Supplementary Fig. 4e), a best-in-class therapeutic agent in this model[20]. Tumor-targeted inhibition of TAP also inhibited the growth of tamoxifen-induced tumors in the BRAF/PTEN melanoma model when used in combination with PD-1 Ab and Flt3 ligand (Fig. 3f).

We next compared the antitumor effect of Nucl-TAP induction of antigenic determinants to vaccination against a mixture of three prototypic mutation-induced neoantigens expressed in the murine MC38 tumor that were identified by exome sequencing, mass spectrometry, and functional validation in mice[21]. As shown in Fig. 3g, vaccination with the peptide mixture and adjuvant was comparable to inducing new antigens by systemic administration of Nucl-TAP. Supplementary Fig. 3f shows the same data as displayed in ref. [21], showing that in both studies the magnitude of tumor inhibition by the peptide vaccine was comparable. Thus, tumor-targeted TAP downregulation enhanced the antigenicity of tumor cells in terms of their ability to stimulate an antitumor immune response (compare "Untreated" or "Nucl-Ctrl" treated to "Nucl-TAP" treated groups), and it was not less potent than vaccinating against spontaneously-arising, tumor cell-expressed, personalized neoantigens. Taken together, the immunotherapy experiments shown in Fig. 3 support the view that the antigen content of tumor cells is an important determinant of tumor

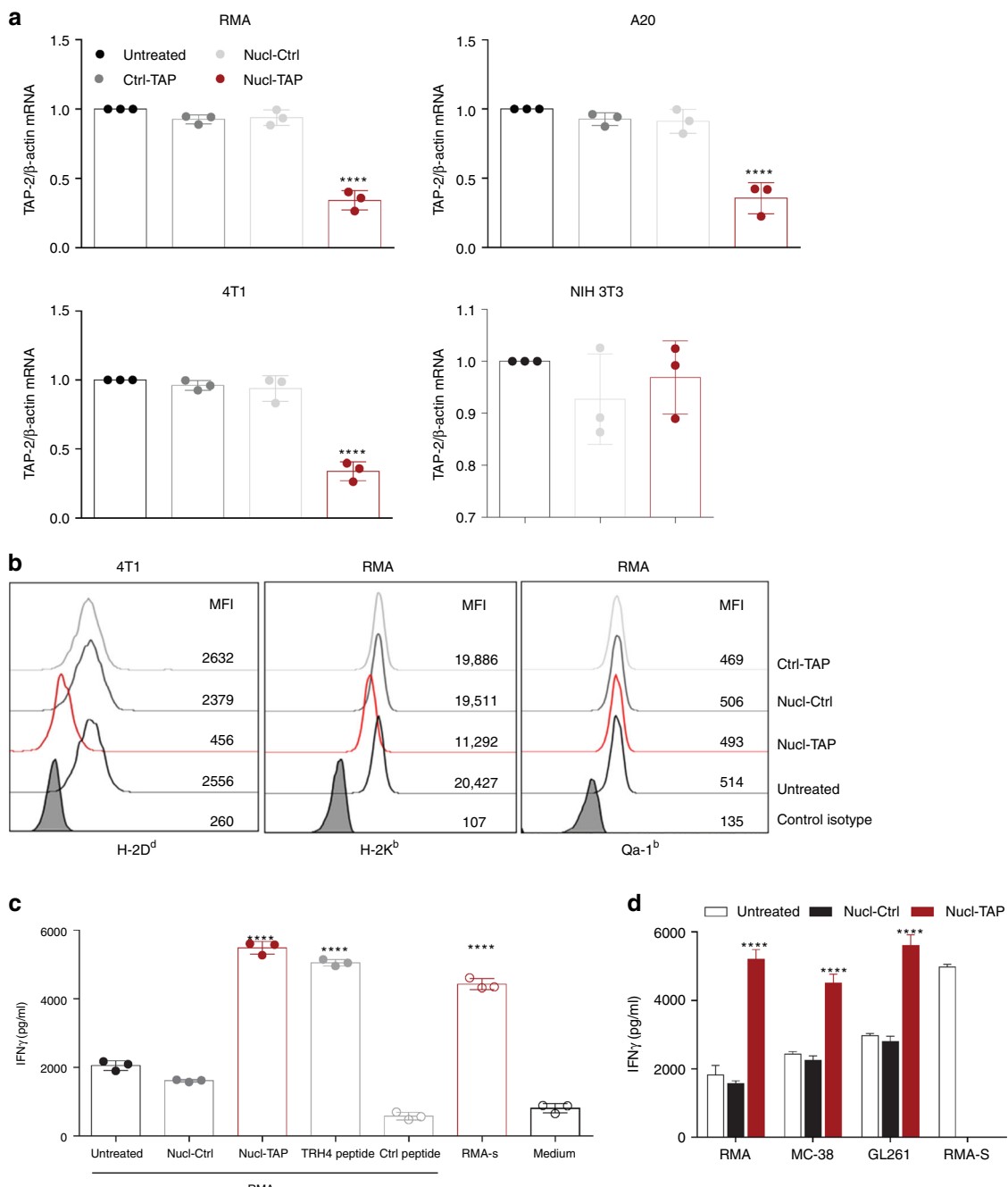

**Fig. 1** Nucleolin aptamer-targeted delivery of TAP siRNA in vitro. **a** Downregulation of TAP RNA in tumor, but not NIH 3T3, cells incubated with Nucl-TAP. Cells were treated with aptamer-siRNAs and 24 h later RNA was isolated and quantified by qRT-PCR. Data represent means and SEM performed in triplicates (*n* = 3). **b** Downregulation of MHC class Ia but not Ib alleles analyzed by flow cytometry in cells as described in panel **a** after 48 h (*n* = 3). **c** and **d** Presentation of the TRH4 peptide. Cells were treated with Nucl-siRNAs or pulsed with peptides and cultured with LnB5 T cells. INFγ production after 20 h stimulation was measured by ELISA. Means and SEM of triplicate wells are represented (*n* = 3). Statistical analyses using one-way ANOVA test and Dunnet posttest for comparison between untreated and treated cells. Differences are indicated: ****$P < 0.001$

immunogenicity[1,2], and that transiently increasing the antigenicity of tumor cells in situ by targeted downregulation of TAP could represent an effective way of potentiating antitumor immunity.

**TAP downregulation induced immune responses in mice.** To determine that tumor inhibition resulting from systemic delivery of Nucl-TAP siRNA was due to TAP downregulation as described for TAP-deficient tumor cells[22], we analyzed key parameters of

the immune response. Antibody depletion experiments in 4T1-bearing mice treated with Nucl-TAP have shown that tumor inhibition was dependent on CD4+, CD8+, as well as on NK cells (Fig. 4a), the latter likely due to the reduced MHC class Ia expression resulting from TAP downregulation (Fig. 2b–d). NK cells can inhibit tumor cells directly by killing the tumor cells, and/or indirectly by recruiting DC to the tumor lesion to promote an adaptive T cell response[23]. Consistent with the latter, in Nucl-TAP-treated mice, DC and NK cells infiltrated tumor lesion at early time points (Fig. 4b, c, day 9) that preceded the appearance

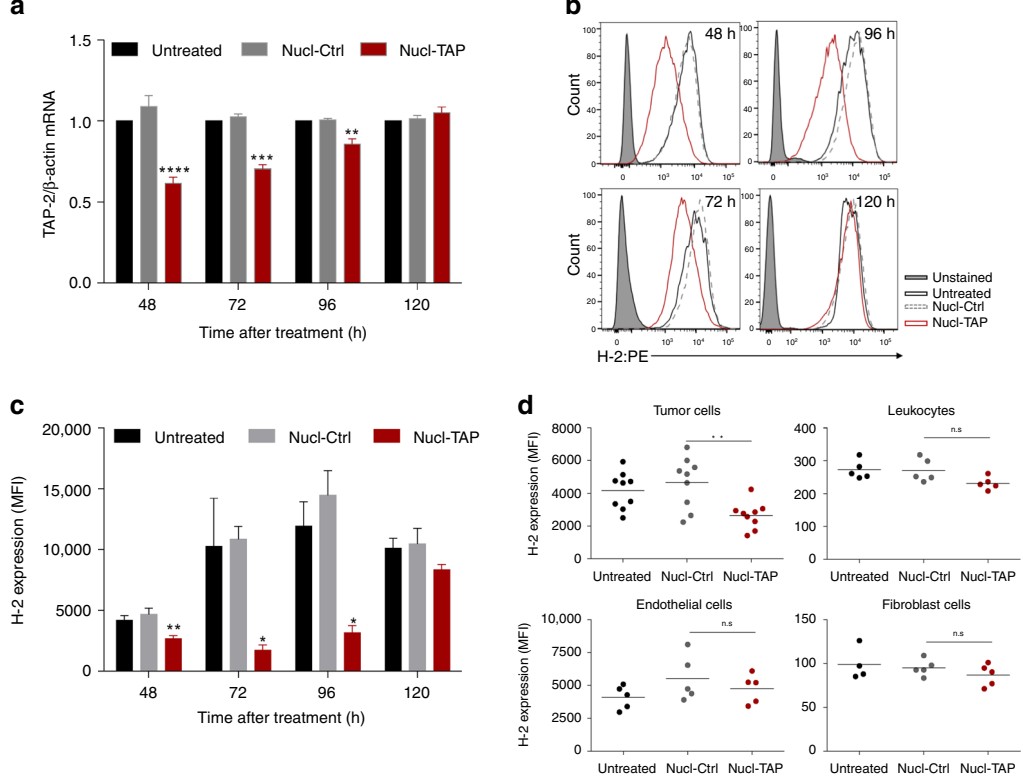

**Fig. 2** Efficiency and specificity of nucleolin aptamer-targeted delivery of TAP siRNA to tumor cells in vivo. Mice bearing subcutaneously implanted A20-GFP tumors were injected once intraperitoneally with Nucl-siRNAs, tumors were excised at the indicated times and analyzed for TAP mRNA and MHC class Ia expression. **a** GFP-expressing cells were isolated by cell sorting and TAP mRNA quantified by qRT-PCR (3 mice/group) ($n = 2$). Data represent means and SEM. **b–d** MHC class Ia expression in A20-GFP tumors treated with Nucl-siRNA conjugates. **b** MHC class Ia expression in GFP+ tumor cells isolated at the indicated times following Nucl-siRNA treatment, representative tumor. **c** Distribution within a group of 3–9 mice represented as mean + SEM. **a** and **c** Statistical analyses using one-way ANOVA test and Dunnet posttest for comparison between untreated and treated cells. **d** MHC class Ia expression 48 h following Nucl-siRNA treatment in GFP+ tumor cells and GFP− tumor resident leukocytes, endothelial cells and fibroblasts (4–9 mice/group). Each circle represents an individual mouse, and means per group are shown ($n = 2$). Statistical analyses using Kruskal–Wallis test and Dunn posttest. Differences are indicated: ****$P < 0.001$, ***$P < 0.005$, **$P < 0.01$, *$P < 0.05$

of CD8+ T cells (Fig. 4b, c, day 12) and the tumor infiltrating NK cells exhibited a mature phenotype at day 9 and activated phenotype at day 12 (Supplementary Fig. 4c). Furthermore, day 9 tumors secreted the NK-derived DC chemokines XCL-1 and CCL5[23] (Fig. 4d). Treatment of tumor-bearing mice with Nucl-TAP was accompanied by a proinflammatory response at the tumor site. In addition to increased numbers of DC, NK cells, and CD8+ T cells (Fig. 4b, c), there was an increased numbers of CD4+ T cells, reduced numbers of foxp3+Treg and Tim3+ PD1+exhausted CD8+ T cells, as well as an increase in the CD8+/Treg ratio (Fig. 4e).

We next evaluated the role of CD8+ T cells. RMA tumor-bearing mice treated with Nucl-TAP elicited T cell responses against the TAP deficiency-induced MHC class I restricted TRH4 epitope measured with tetramers (Fig. 4f) or using an in vivo cytotoxicity assay (Fig. 4g). Consistent with the shared and polyclonal nature of TAP downregulation-induced antigens, CD8+ T cells isolated from Nucl-TAP treated MC38 tumor-bearing mice recognized in vitro Nucl-TAP-treated RMA tumor cells (Fig. 4h), DC2.4 cells pulsed with the Kb-restricted TRH4 peptide as well as the TAP downregulation-induced Qa-1b-restricted FAP peptide[24] (Fig. 4i), and when transferred to mice inhibited the growth of TAP-deficient RMA-S tumor cells (Fig. 4j). Epitope spreading is thought to be a hallmark of an increasingly potent immune response[25]. CD8+ T cells isolated from RMA tumor-bearing mice that were treated with Nucl-TAP

or Nucl-Ctrl siRNA were transferred to RMA tumor-bearing mice that were or were not treated with Nucl-TAP siRNA. Consistent with epitope spreading, adoptively transferred CD8+ T cells from Nucl-TAP, but not Nucl-Ctrl, siRNA-treated mice inhibited tumor growth in tumor-bearing mice that were not treated with Nucl-TAP siRNA, and was further enhanced upon Nucl-TAP siRNA treatment (Fig. 4k). Similarly, in a second model, CD8+ T cells isolated from 67NR breast carcinoma tumor-bearing mice treated with Nucl-TAP, but not Nucl-Ctrl, siRNAs inhibited the growth of 67NR tumor cells though to a lesser extent than 67NR tumor-bearing mice treated with Nucl-TAP siRNA (Fig. 4l). Taken together, these experiments confirm the T cell-mediated immunological basis of Nucl-TAP-mediated tumor inhibition, but also underscore the importance of NK cells, showing that TAP downregulation in tumor cells promotes the adaptive, as well as the innate arm of the antitumor immune response.

**TAP downregulation does not elicit autoimmune pathologies.** Administration of the nucleolin aptamer AS1411 used in our study was found to be safe in cancer patients used at 10–100 fold higher doses/kg[17] compared to the doses used to treat mice with the Nucl-siRNA conjugates. However, presentation of the normally cryptic TAP deficiency-induced epitopes, or nonspecific uptake of TAP siRNA by normal cells, could cause autoimmune pathology. No evidence of toxicity was observed in 4T1 tumor-bearing mice treated six times with Nucl-TAP in terms of changes

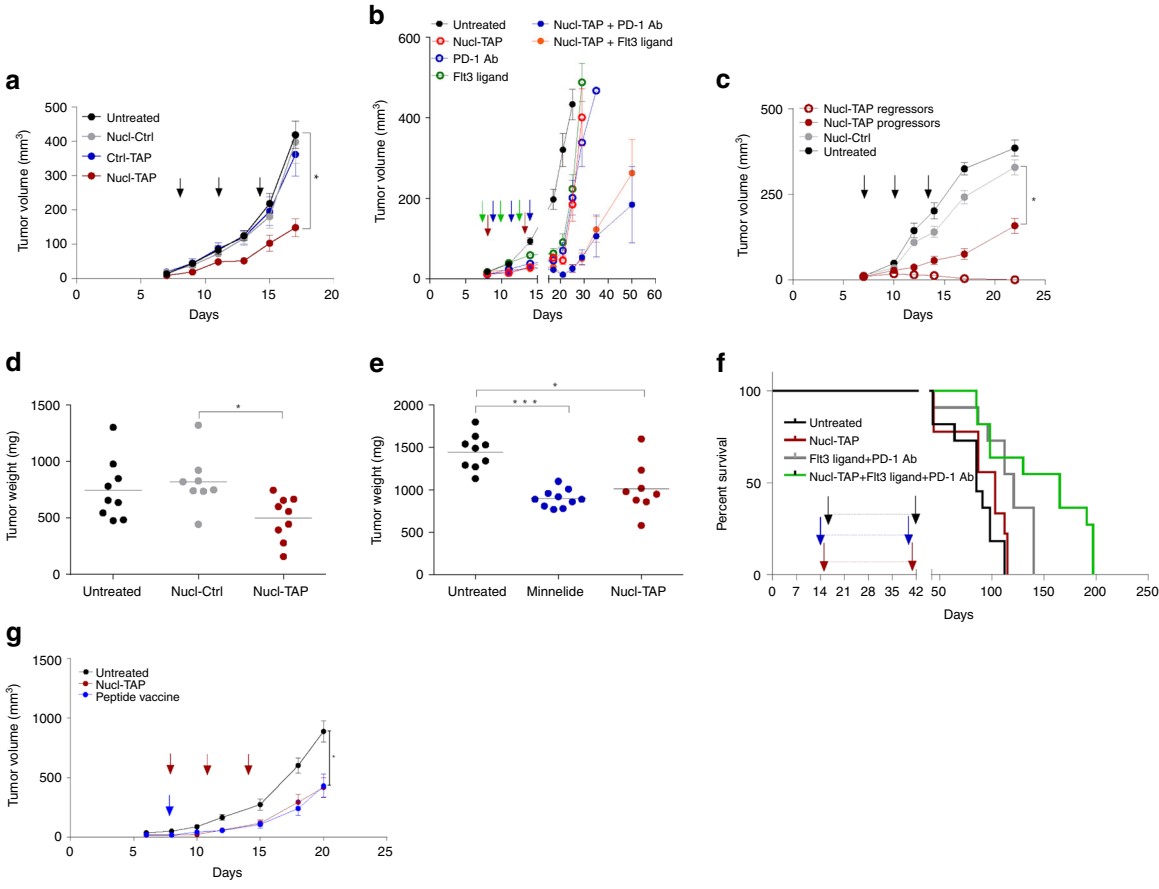

**Fig. 3** Nucleolin aptamer-targeted TAP siRNA-mediated inhibition of tumor growth in mice. **a** 4T1 breast carcinoma model. Arrows: treatment with aptamer-siRNA conjugate (11 mice/group) ($n = 3$). **b** As in panel **a** except that the 4T1 tumor-bearing mice were treated only twice instead of three times with Nucl-TAP. Where indicated, mice were treated with PD-1 Ab or with Flt3 ligand. Arrows: Nucl-TAP, red; PD-1 Ab, blue or Flt3 ligand, green (9–10 mice/group) ($n = 2$). Nucl-TAP versus Nucl-TAP+PD-1 Ab, $P < 0.0001$; PD-1 Ab versus Nucl-TAP+PD-1 Ab, $P = 0.0003$; Nucl-TAP versus Nucl-TAP+Flt3 ligand, $P < 0.0001$; Flt3 ligand versus Nucl-TAP+Flt3 ligand, $P = 0.0238$. **c** A20 B lymphoma model. Arrows: treatment with aptamer-siRNA conjugates (untreated and Nucl-Ctrl, 7 mice/group; Nucl-TAP, 30 mice/group) ($n = 2$). Progressors (19 mice), regressors (11 mice). For **a**, **b**, and **c**, data represent means and SEM. **d** and **e** Orthotopic KPC-derived pancreatic cancer model (experimental protocol shown in Supplementary Fig 3c). **d** Tumor weight 37 days post tumor implantation (8–9 mice/group) ($n = 2$). **e** Same as **d** except for including a group of mice treated with Minnelide (9–10 mice/group) ($n = 1$). For **d** and **e**, each circle represent an individual mouse, and means per group are shown. **f** BRAF/PTEN melanoma model. Tumor-bearing mice treated with a combination of Nucl-TAP, PD-1 Ab, and Ftl3 ligand. Arrows indicated beginning and end of treatment. Red, Nucl-TAP; black, PD-1 Ab; blue; Flt3 ligand. Untreated versus Nucl-TAP, $p = 0.2657$; Untreated versus PD-1 Ab+Flt3 ligand, $P = 0.1168$; Untreated versus combination 3×, $p = 0.0067$; Nucl-TAP versus combination 3×, $P = 0.0133$; PD-1 Ab+Flt3 ligand versus combination 3x $P = 0.0543$ ($n = 1$). **g** MC38 adenocarcinoma model. MC38 tumor-bearing mice were treated with either Nucl-TAP or a mixture of 3 peptides mixed in adjuvant as described in[21]. Arrows: Nucl-TAP, red; Peptide vaccine, blue. Two experiments performed are shown together (11–14 mice/group). Data represent means and SEM. **a–c** and **g** statistical analyses using one-way ANOVA test and Tuckey posttest. **d** and **e** statistical analyses using Kruskal–Wallis test and Dunn posttest. **f** Survival curves analyzed by Log-rank (Mantel–Cox) test. **a**, **c**, **d**, **e**, and **g** differences are indicated in graphs: ***$P < 0.005$, *$P < 0.05$

in organ weight (Fig. 5a), CD4$^+$/CD8$^+$ cell ratio in the spleen (Fig. 5b), activation markers of CD4$^+$ and CD8$^+$ T cells (Fig. 5c), hematology parameters (Supplementary Table 1), serum cytokines, despite considerable mouse-to-mouse variation (Fig. 5d), or liver damage as determined by measuring alanine transaminase (ALT) or aspartate transaminase (AST) levels in the circulation (Fig. 5e). There was also no change seen after three administrations of Nucl-TAP, alone or in combination with PD-1 Ab or Flt3 ligand, in term of organ weight, except for an expected increase in the weight of spleens and lymph nodes of mice treated with Flt3 ligand (Supplementary Fig. 5a), and ALT or AST levels in the circulation (Supplementary Fig. 5b). Importantly, no inflammation was seen in the small intestine, liver, or lung, in contrast to treatment with a comparably therapeutic dose of CTLA-4 Ab[26] that elicited organ-wide inflammatory responses often seen in patients that respond to this antibody[27] (Fig. 5f and

Supplementary Fig. 5c). Taken together, these experiments suggest that systemic administration of a therapeutic dose of Nucl-TAP does not elicit short-term toxicities in mice and exhibits a therapeutic index that is superior to that of CTLA-4 Ab. However, the possibility that repeated administrations of Nucl-TAP over an extended period time will induce autoimmune pathologies remains to be determined.

**Presentation of a common epitope in human tumor cells**. To test if this approach could be applicable to human patients we generated a human TAP siRNA that was conjugated to the nucleolin aptamer. Human tumor cell lines of melanoma and renal origin[28] incubated with Nucl-TAP downregulated TAP mRNA (Fig. 6a), and HLA expression (Fig. 6b, c). Marijt et al.[28] have shown that tumor cells in which TAP expression is knocked

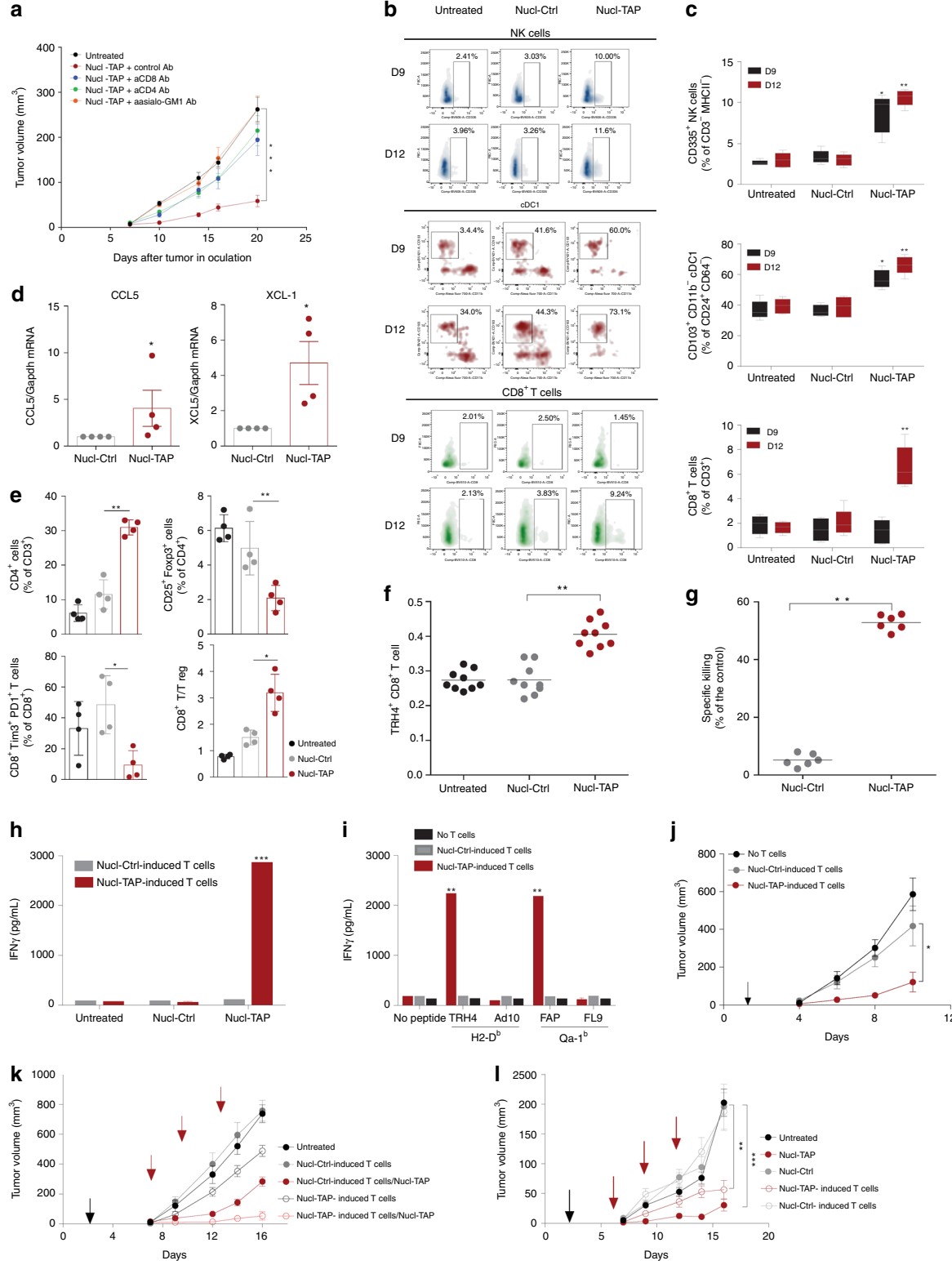

out by CRISPR/CAS9 present a new HLA-A2-restricted peptide derived from the housekeeping LRPAP1 protein that was capable of activating a T cell clone specific to the TAP deficiency-induced HLA-A2-restricted peptide. As shown in Fig. 6d, e, treatment of melanoma or renal cancer tumor cells with Nucl-TAP led to the activation of lentiviral transduced CD8+ T cells expressing the TCR specific to LRPAP1 peptide as measured by the acquisition of cytotoxic (Fig. 6d) and IFNγ-secreting (Fig. 6e) effector functions.

## Discussion

In this study, we have described a simple and clinically applicable method to enhance the antigenicity of tumor cells in situ by downregulation of TAP expression in tumor cells. We have shown that tumor-targeted siRNA-mediated downregulation of TAP inhibits tumor growth in transplantable, orthotopic, and autochthonous murine tumor models of distinct origin and genetic background (Fig. 3, Supplementary Fig. 3). As shown in Fig. 3b, f, combination of TAP inhibition with checkpoint

**Fig. 4** Immune responses elicited in mice treated with Nucl-TAP. **a** CD4[+] T cell, CD8[+] T cell, and NK cells dependence of tumor inhibition. 4T1 tumor-bearing mice treated with Nucl-TAP, and anti-CD4, anti-CD8, or anti-asialo-GM1 antibodies (6 mice/group). Data represent means and SEM ($n = 2$). **b** and **c** Accumulation of cDC1, NK and CD8+ T cells at day 9 and day 12 in 4T1 tumors after the first and second dose of Nucl-TAP siRNA administration, respectively. **b** Individual mice represented as density plots. **c** Distribution within a group of four mice represented as box plot analysis (see gating strategy in Supplementary Fig. 4a, b). **d** Secretion of XCL-1 and CCL5 in day 9 4T1 tumors. Data represent means and SEM. **e** Tumor-infiltrating immune subsets. 4T1 tumors were resected from mice 2 days after second dose of Nucl-TAP and immune cells subsets were analyzed by flow cytometry. Data represent mean and SEM (4 mice/group) ($n = 2$). **f** and **g** Treatment of RMA tumor-bearing mice with Nucl-TAP elicits TRH4-specific CD8[+] T cell responses. **f** Staining with TRH4-H-2D[b] tetramer on tumor-infiltrating lymphocytes 2 days after second dose administration (9 mice/group) ($n = 2$). **g** In vivo killing of TRH4 peptide pulsed splenocytes as described in the "Methods" section. Each circle represents an individual mouse, and means per group are shown. **h** and **i** CD8[+] T cells isolated from MC38 tumor-bearing mice treated with Nucl-TAP recognize in vitro RMA tumor cells incubated with Nucl-TAP **h** or DC2.4 cells pulsed with the TAP downregulation induced Kb-restricted TRH4 peptide and Qa-1b restricted FAP peptide **i**. IFNγ production after 20 h stimulation was measured by ELISA. Means and SEM of triplicate wells are represented ($n = 2$). **j** Adoptive transfer of TAP-specific CD8[+] T cells from MC38 bearing mice treated with Nucl-TAP inhibits the growth of the TAP-deficient RMA-S tumors. Arrows: T cells infusion (7 mice/group). Data represent means and SEM ($n = 1$). **k** and **l** TAP downregulation mediated induction of non-TAP deficient specific CD8[+] T antitumor immunity. Adoptive transfer of CD8[+] T cells from RMA **k** or 67NR **l** bearing mice treated with Nucl-TAP inhibits the growth of these tumors in the absence of Nucl-TAP treatment. Black arrow, T cells infusion: red arrows, Nucl-siRNAs treatment (6–9 mice/group). Data represent means and SEM ($n = 1$). For **k**, statistical analysis at day 16: Nucl-Ctrl-induced T cells versus Nucl-TAP-induced T cells, $P = 0.0060$; Nucl-Ctrl-induced T cells versus Nucl-Ctrl-induced T cells/Nucl-TAP, $p < 0.0001$; Nucl-Ctrl-induced T cells versus Nucl-TAP-induced T cells/Nucl-TAP, $P < 0.0001$; Nucl-TAP-induced T cells versus Nucl-Ctrl-induced T cells/Nucl-TAP, n.s; Nucl-Ctrl-induced T cells/Nucl-TAP versus Nucl-TAP-induced T cells/Nucl-TAP, n.s; Nucl-TAP-induced T cells versus Nucl-TAP-induced T cells/Nucl-TAP; $P < 0.0001$. **a, c, e, h–l** Statistical analyses using one-way ANOVA test and Tuckey posttest. **d** Statistical analyses using Mann–Whitney test. **f** and **g** Statistical analyses using Kruskal–Wallis and Dunn posttest. **c, h,** and **l** Comparisons between Nucl-Ctrl versus Nucl-TAP-treated/induced cells. **a, c–j, l** differences are indicated in graphs: ***$P < 0.005$, **$P < 0.01$, *$P < 0.05$

blockade using PD-1 Ab and/or recruiting DC to the tumor by treatment with Flt3 ligand, further potentiated the antitumor response, suggesting that increasing the antigenicity of tumor cells in situ by transient TAP downregulation could enhance the effectiveness of any immune-potentiating therapy, including but not limited to checkpoint blockade.

Underscored by recent clinical trials[29], responsiveness to checkpoint blockade was shown to correlate with the number of nonsynonymous mutations in the patient' tumors, a subset of which are neoantigens thought to elicit effective antitumor immunity[5,6,30]. Since many tumors do not express or express too few neoantigens[3,4], the majority of patients are less responsive to checkpoint blockade therapy[5,6], and conceivably to other forms of immune therapy. The ability to induce the expression of new antigens in the patients' tumors that is capable of eliciting potent antitumor immunity in the absence of significant toxicity as our study suggests, will broaden the responsiveness of patient to checkpoint blockade and other form of immune therapy, including patients that do not express or express too few endogenous neoantigens[3,4].

Recent studies underscoring the immense variability among the patient' metastatic lesions also in terms of antigenic composition, and the clinical importance of least-infiltrated metastatic lesion(s)[31,32], suggest that the neoantigens identified in accessible biopsies that do not correspond to the least-infiltrated, often disseminated and hence hard to identify, metastatic lesions, may have limited therapeutic benefit. Our approach, whereby the Nucl-TAP siRNA is administered systemically in the circulation, will ensure that the induced antigens will be also expressed in the critical least-infiltrated disseminated lesions.

Consistent with the hypothesis that TAP downregulation-induced antigens are potent tumor rejection antigens, in MC38 tumor-bearing mice inhibition of TAP was comparatively effective to vaccination with a mixture of prototypic clonal neoantigens[21](Fig. 3g and Supplementary Fig. 3f), suggesting that the tumor cells, by virtue of their increased antigenicity, were capable of stimulating an endogenous immune response that was comparable to that of an immune response elicited by vaccinating against endogenous neoantigens. Furthermore, in contrast to the need of identifying neoantigens on a patient-by-patient basis, which is yet to be perfected, tumor-targeted TAP downregulation

is achieved using a systemically administered, chemically synthesized reagent that would be applicable to most, if not all cancer patients. Since the antigenic epitopes induced upon TAP downregulation are encoded in housekeeping products[22], the same epitopes will be presented in all (tumor) cells in which TAP was downregulated, as was indeed shown in TAP-deficient tumor cells[28,33] as well as in tumor cells treated with TAP siRNA (Figs. 1d, 4h, j and 6d, e). Such epitopes are the equivalent of clonal neoantigens exemplified by the three MC38 tumor-encoded neoantigens used in the comparative studies (Fig. 3g). However, since clonal neoantigens represent a small proportion of the patients' neoantigenic repertoire their identification is likely to be especially challenging. Furthermore, neoantigens, especially of the clonal type that have been exposed to the immune system for the longest time, have elaborated multiple mechanisms of immune evasion[16,34–37]. It is conceivable that since siRNA-mediated expression of the TAP downregulation-induced epitopes will be transient, the risk of immune evasion will be significantly reduced. Taken together, our studies suggest that inducing antigens in tumor cells in situ by nucleolin-targeted downregulation of TAP could be as if not more effective than vaccinating against endogenous neoantigens, clinically more feasible, and broadly applicable to all cancer patients.

Tumor targeting of therapeutics, including siRNAs, has been limited by the paucity of tumor-specific targets. Nucleolin could potentially serve as a broadly useful target to deliver therapeutics to tumor cells in vivo, and the nucleolin aptamer developed by Bates and colleagues that binds to both murine and human nucleolin[17] could represent an all but universal tumor-targeting ligand. The nucleolin-binding aptamer was used to target a broad range of biological agents to tumor-bearing mice and human tumor cells in vitro, and was used in clinical trials of cancer patients as a cytotoxic agent, administered at doses that were 10–100 higher than the doses used in our murine studies, without inducing significant toxicities[17]. Using the nucleolin-binding aptamer as a prototype for a tumor targeting ligand, we have shown that targeting the systemically administered nucleolin aptamer-conjugated TAP siRNA (Nucl-TAP) to tumor cells in mice was very efficient and specific (Fig. 2). Underscoring the broad utility of the nucleolin aptamer as tumor-targeting ligand for siRNA delivery, measuring mRNA (Fig. 1a) or MHC class I

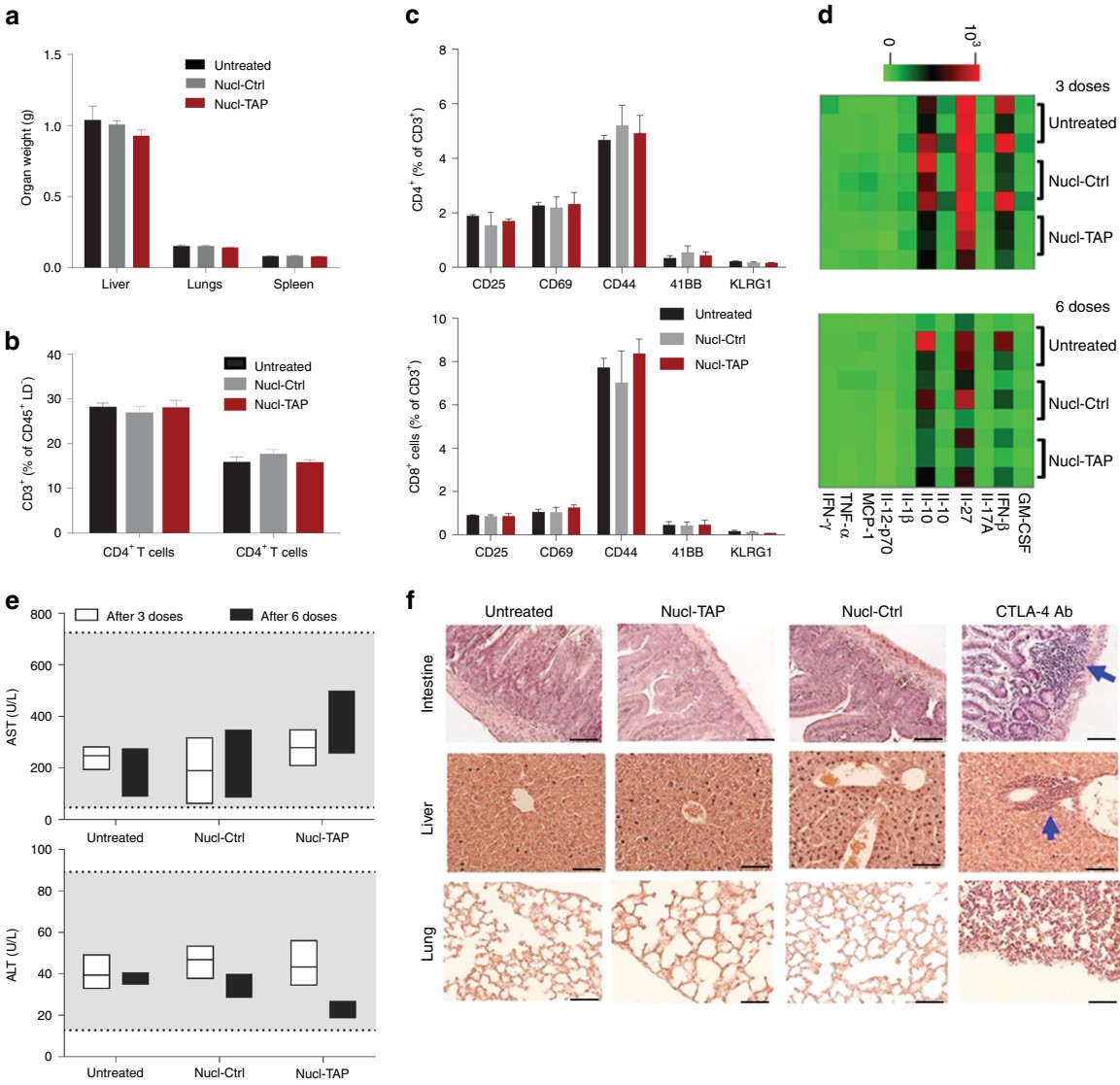

**Fig. 5** Toxicity in mice treated with Nucl-TAP. Balb/c mice (8 mice/group) were administered with Nucl-TAP or Nucl-Ctrl, six times every 3 days, except for the immunohistochemistry analysis (panel **f**) where mice were treated three times. Two days post last administration mice were sacrificed and analyzed. **a** Organ weight. **b** CD4/CD8 T cell ratio in the spleen. **c** Activation markers expressed on CD4$^+$ and CD8$^+$ T cells isolated from the spleen. Data represent means and SEM. **d** Cytokine levels in the serum of mice after three or six administrations of Nucl-TAP. Heatmap shows log-transformed cytokine concentration values for the experimental groups (3 mice/group). **e** Liver pathology. ALT and AST in the circulation were determined by ELISA after three or six administrations of Nucl-TAP. Shaded area represents normal levels of ALT or AST in Balb/c mice (from The Jackson Laboratories (MPD, http://phenome.jax.org/)). Data represent box plot analysis. **f** Inflammatory responses in tissue sections stained with hematoxylin and eosin and visualized by light microscopy at ×40 magnification (scale bar: 25 μm). One group of mice was also treated with 200 μg of CTLA-4 Ab that elicits a comparable antitumor effect[27,41]. Arrows indicate inflammatory foci in mice. **a–c** Statistical analyses using Kruskal–Wallis and Dunn posttest. **e** Statistical analyses using two-way ANOVA

downregulation (Fig. 1b), antigen presentation in vitro (Fig. 1c, d), CTL induction (Fig. 4f–i), or tumor inhibition in mice (Fig. 3), and in vitro studies in human tumor cells (Fig. 6), the TAP siRNA was delivered to multiple murine and human tumor cells of distinct origin.

The mechanism underlying the cancer cell specificity of the nucleolin aptamer-targeted siRNA delivery is not well under-stood. Work from Hovanessian's lab suggests that differential expression of nucleolin on the cell surface of cancer cells results from constitutive cell surface trafficking of nucleolin, compared to mitogen-dependent nucleolin cell surface trafficking in normal cells[38]. A mutually nonexclusive explanation for the differential nucleolin aptamer-conjugated siRNA delivery to cancer cells proposed by Reyes-Reyes et al. is that the uptake of the aptamer

cross-linked nucleolin on the cell surface of cancer, but not normal cells, is mediated via macropinocytosis and hence delivery of the complex to the cytoplasm, whereas in non-transformed cells the aptamer is either recycled or transported to the lysosome[39].

Unlike mutation-induced neoantigens, the TAP downregulation-induced antigens are cryptic antigens encoded in housekeeping products that are presented when TAP is down-regulated[22]. If under some circumstances the otherwise cryptic epitopes are presented by normal tissues, or despite tumor tar-geting some of the siRNA will be taken up by normal cells, autoimmune pathology could develop. Evidence that TAP-deficiency-induced neoepitopes are not presented in mice[10] even under systemic inflammation[40] or in humans[28], was

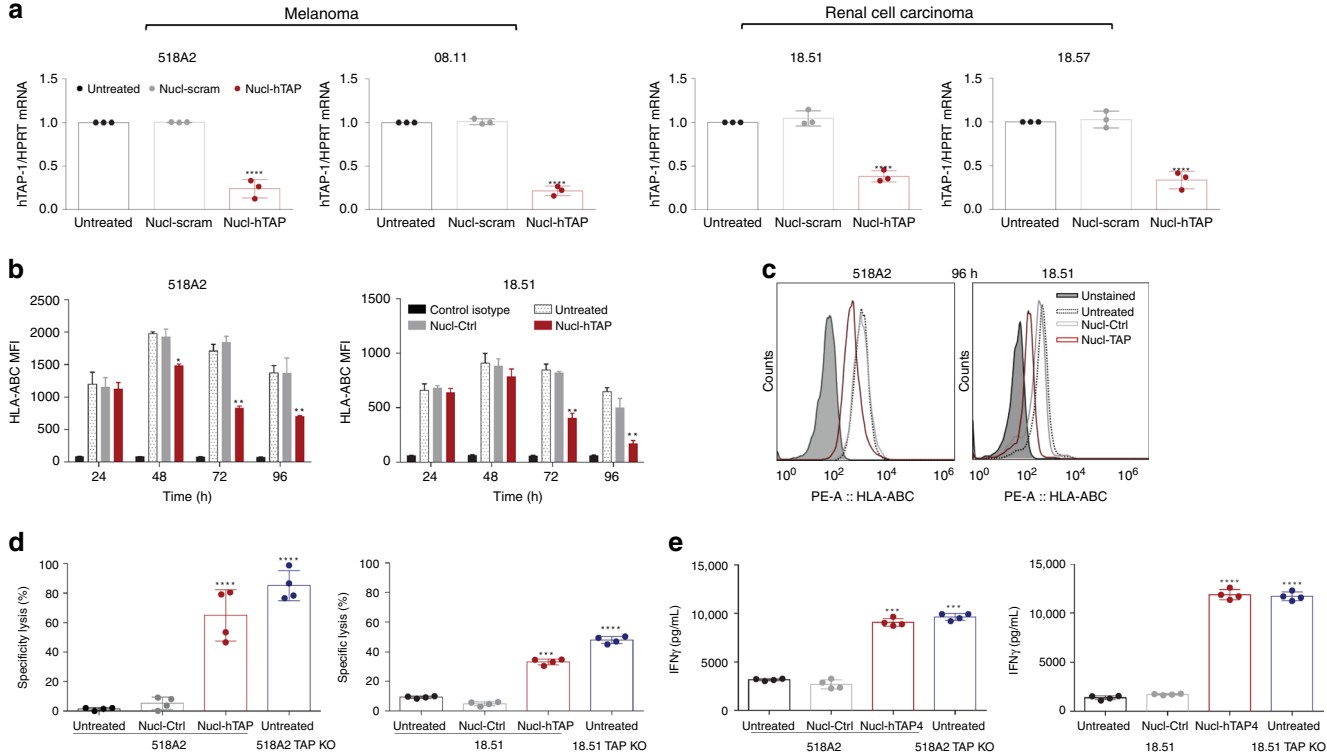

**Fig. 6** Presentation of a TAP-downregulation-induced peptide in human tumors treated with Nucl-TAP. **a** Downregulation of TAP mRNA in human tumor cells incubated with Nucl-TAP. Cells were treated with Nucl-siRNAs and 24 h later mRNA was isolated and quantified by qRT-PCR. Data represent means and SEM performed in triplicates ($n = 2$). **b** and **c** Downregulation of HLA-A, B, and C analyzed by flow cytometry at different times as indicated ($n = -3$). **b** Data represent means and SEM performed in quadruplicates. **c** Representative experimental well at 96 h. **d** and **e** Presentation of a TAP downregulation-induced peptide in TAP-deficient (TAP KO) or TAP-sufficient 518A2 and MZ18.51 tumor cells treated with Nucl-siRNAs for 72 h, and cultured with a CD8[+] T cell clone that recognized the HLA-A2-TAP downregulation-induced peptide complex[29]. **d** In vitro cytotoxicity after 4 h coculture was determine by lactate dehydrogenase assay. **e** INFγ production after 20 h stimulation was measured by ELISA. Means and SEM of quadruplicate wells ($n = 1$). Statistical analyses using one-way ANOVA test and Dunnet posttest for comparison between untreated and treated cells. Differences are indicated: ****$P < 0.001$, ***$P < 0.005$, **$P < 0.01$, *$P < 0.05$

suggested by studies showing that cognate T cells remain naïve. In this study, we have not seen any indication of toxicity in mice treated with nucleolin aptamer-targeted TAP siRNAs (Fig. 5, Supplementary Fig. 5, and Supplementary Table 1), whereas a comparatively therapeutic dose of CTLA-4 Ab[41]-elicited non-specific inflammatory responses in various organs (Fig. 5f and Supplementary Fig. 5c), similar to what has been seen in human patients[27]. Thus, while our studies have not seen evidence of toxicity in mice treated with therapeutic doses of Nucl-TAP, only clinical trials will be able to definitively address this concern.

The mechanism underlying the generation of TAP-independent epitopes that are restricted not only by the canonical polymorphic MHC class Ia alleles but also by the oligo-morphic HLA-E/Qa-1 alleles is well established[22]. Nucl-TAP inhibition of tumor growth in mice was dependent on CD8[+] T cells as shown by Ab depletion (Fig. 4a) and adoptive transfer (Fig. 4j) experiments. Furthermore, incubation of tumor cells in vitro with Nucl-TAP led to the presentation of a previously described TAP downregulation specific K[b]-restricted epitope (Fig. 1c), and CD8[+] T cells isolated from tumor-bearing mice treated with Nucl-TAP, but not Nucl-Ctrl, siRNA recognized DC pulsed with either an K[b] or Qa-1[b] restricted TAP downregulation-induced peptide (Fig. 4i). Whether enhanced antitumor CD8[+] T cell immunity was mediated by direct and/or cross-presentation of the TAP downregulation-induced antigens by the tumor cells themselves or by tumor-resident APC, respectively, remains to be determined. Evidence of epitope spread in two models (Fig. 4k, l) is suggestive of the potency of

the antitumor immune response elicited against the experimentally induced antigens[25], and could control tumor growth even if TAP is downregulated in a proportion of the tumor cells.

Nucl-TAP inhibition of tumor growth in mice was dependent on NK cells (Fig. 4a). In mice and humans TAP deficiency is generally associated with increased tumorigenicity[42–44], conceivably representing an immune evasion mechanism due to an overall reduction of antigen presentation to CD8[+] T cells by the polymorphic MHC class Ia molecules that are partially down-regulated when TAP expression is reduced. In a few studies, however, TAP deficiency was associated with reduced tumorigenicity[45,46]. These apparently conflicting observations may be due to differential susceptibility of TAP-deficient tumor cells to NK cells caused by the reduced MHC class Ia expression[22] that were seen also in our studies (Fig. 1b).

The knockdown of TAP in tumor cells to enhance tumor cell immunogenicity seems counterintuitive given that this is a potential immune evasion mechanism employed by tumor cells[44] and some pathogens[47]. As previously described, and as our results confirm, TAP downregulation leads to transient, though partial, downregulation of TAP MHC class Ia expression on the cell surface (Figs. 1b and 2b–d). This is probably a consequence of the absence of the TAP-transported peptides that otherwise stabilize the nascent MHC molecules, replaced by TAP-independent housekeeping products derived peptides present at a lower concentration. Thus in TAP-deficient cells a smaller repertoire of peptides encoded in secreted products gain access to the ER and associate not only with nascent MHC class Ia molecules but also

with the semi-invariant Qa-1/HLA-E molecules (in the latter because of reduced MHC leader peptide generation). Consequently, TAP deficiency induces a new set of peptides presented by both canonical and semi-invariant MHC molecules, but which are not presented by the TAP-sufficient cells, thereby defining the antigenicity of TAP-deficient, but not TAP-sufficient, cells. Why the TAP-deficient tumor cells in cancer patients predicted to express new TAP-independent antigens do not elicit a protective immune response? A likely explanation is the absence of costimulatory molecules on tumor cells, inefficient cross-presentation to professional antigen-presenting cells, and/or the immune suppressive environment in the tumor milieu. Our study shows that transient expression of TAP downregulation-induced antigens in the tumor cells was able to shift the balance above a threshold to elicit a measurable protective antitumor immune response in multiple tumor models (Fig. 3). Conceivably, this reflects the potency of the induced antigens and the effectiveness of the antigen induction process with the Nucl-TAP siRNA conjugate.

Human tumor cells in which TAP is experimentally[28,48] or naturally downregulated also present novel class I-restricted peptides that can stimulate T cells in vitro[28,48] or in vivo[49], respectively. Analysis of one such HLA-A2-restricted peptide encoded in the housekeeping LRPAP1 protein has shown that it is presented by multiple HLA-A2-positive TAP-deficient tumor cells of distinct origin[28]. We have shown that Nucl-TAP-mediated TAP downregulation in melanoma and renal tumor cell lines downregulated TAP mRNA (Fig. 6a) and HLA expression (Fig. 6b, c), and induced the presentation of the previously described LRPAP1-derived peptide that was recognized by CD8$^+$ T cells transduced with the corresponding TCR-expressing lentivirus (Fig. 6d, e). These observations support the view that downregulation of TAP in human tumor cells will also induce the presentation of shared new epitopes in the patient' tumors, the equivalent of the rare and difficult-to-identify patient-specific clonal neoantigens[1,2].

TAP is a prototype of a new class of antigen-inducing targets comprising of mediators of antigens processing and presentation, such as ERAAP[50] or Invariant chain (Ii)[51]. For example, we found that nucleolin-targeted siRNA inhibition of ERAAP also engendered antitumor immunity in the transplantable 4T1 breast carcinoma model (Supplementary Fig. 3b). Future studies will explore downregulation of which (combination of) mediators will elicit the most effective antitumor response.

In summary, our study describes a simple and broadly applicable approach to increase the antigenicity of tumor cells in situ that could potentiate the responsiveness of patient to immune therapy, especially patient with otherwise low intrinsic mutation burden[3,5,6].

## Methods

**Mouse strains**. All animal work was conducted under the approval of the University of Miami Institutional Animal Care and Use Committee (IACUC) in accordance with federal, state and local guidelines. C57BL/6 and BALB/c mice were purchased from The Jackson Laboratories. Braf/Pten (B6.Cg-Braf$^{tm1Mmcm}$P-ten$^{tm1Hwu}$Tg(Tyr-CreER$^{T2}$)13Bos/BosJ) and Pten$^{lox/lox}$ mice were kindly provided by Dr. Marcus Bossenberg. All Braf/Pten mice were genotyped, and deletions confirmed by PCR as previously described[52].

**Tumor models**. 4T1 breast carcinoma model: 7–9-week-old female Balb/c mice were injected subcutaneously with $2 \times 10^4$ 4T1 tumor cells. Eight days after tumor inoculation (palpable tumors with volume of ~25–50 mm³), Nucl-siRNAs were administered i.p. at 1 nmol dose (1.75 mg/kg), repeated two additional times 3 days apart. For the analysis of immune cell infiltrates, tumors were resected from mice 2 days after the second dose and processed as described below. Cellular subsets were depleted by administering depleting antibody i.p. twice weekly beginning 6 days before tumor implantation: CD8$^+$ T cells with anti-CD8-α (200 μg, clone 2.43, BioXCell), CD4$^+$ T cells with anti-CD4 (200 μg, clone GK1.5, BioXCell), NK

cells with anti-asialo GM1 (20 μl, clone Poly21460, Biolegend). Cellular depletions of CD8$^+$ T cells, CD4$^+$ T cells, and NK cells were confirmed by flow cytometry. (Of note, the expression of asialo-GM1 is not strictly confined to NK cells among hematopoietic cells and is detected on a subpopulation of NKT, CD8$^+$ T, and γδ T cells and some activated form of CD4$^+$ T cells, macrophages, and eosinophils under certain experimental conditions). For combination experiments, mice were treated i.p. with either 200 μg of PD-1 Ab clone RMPI-14 (BioXCell) one day after each Nucl-TAP or –ERAAP siRNA injection, or with 20 μg of Flt3 ligand (BioX-Cell) one day before each Nucl-TAP siRNA injection. For the combination experiments, mice were treated only twice with Nucl-TAP siRNA. As positive control of systemic inflammation, mice were injected with 200 μg of CTLA4 Ab clone 9H10 (BioXcell) as described previously[41].

67NR breast carcinoma model: 7–9-week-old female Balb/c mice were injected subcutaneously with $1 \times 10^5$ 67NR tumor cells. Seven days after tumor inoculation (palpable tumors with volume of ~5–40 mm³) treatment was initiated. Nucl-siRNA treatment schedule and dose were the same as for the 4T1 model. For adoptive cell transfer experiments, 67NR-bearing mice received one infusion of CD8$^+$ T cells ($0.25 \times 10^6$) 2 days after tumor implantation. For the generation of TAP-deficient specific CD8$^+$ T cells, 67NR-bearing mice that have received two doses of Nucl-siRNA conjugates were euthanized 2 days after the second dose. Cells from tumor-draining lymph nodes were isolated and restimulated in vitro during 5 days with IL-2 (20 IU/ml) in the presence of irradiated TAP or control RNA-expressing D2SC1 DC cell line (1:3, APC:target ratio) and autologous splenocytes (2.5:1, splenocytes:target ratio). CD8$^+$ T cells were purified using a MACS-negative selection column (Miltenyi Biotec).

A20 B lymphoma model: 7–9-week-old female Balb/c mice were injected s.c. with $1 \times 10^6$ A20 tumor cells and 6–7 days after inoculation (palpable tumors with volume of ~10–25 mm³) treatment was initiated. Treatment schedule and dose were the same as for the 4T1 model. For testing efficiency of nucleolin-targeted TAP siRNA delivery in vivo, Balb/c mice were injected subcutaneously with $1 \times 10^6$ GFP-expressing A20 tumor cells. Ten days after tumor inoculation (150 mm³ as tumor volume average), mice were treated once with Nucl-siRNAs, and 24, 48, 72, and 96 h later tumors were harvested and processed for flow cytometry or cell sorting.

RMA T lymphoma model: 7–9-week-old female C57BL/6 mice were injected s.c. with $5 \times 10^4$ RMA tumor cells and 6–7 days after inoculation (palpable tumors with volume of ~10–25 mm³) treatment with Nucl-TAP siRNA was initiated. Treatment schedule and dose were the same as for the 4T1 model. For in vivo cytotoxicity assay, syngeneic naive splenocytes were isolated and labeled with either 5 μM CFSE (CFSE$^{hi}$ cells) or 0.5 μM CFSE (CFSE$^{lo}$ cells). CFSE$^{hi}$ cells were pulsed with THR4 peptide, and CFSE$^{lo}$ cells were pulsed with an irrelevant peptide for H-2D$^b$ (Ad10, SGPSNTPPEI)[13]. Cells were then injected i.v. in a 1:1 ratio in RMA-tumor-bearing mice treated with Nucl-siRNAs or control. Forty-eight hours later, spleens were harvested and CFSE-labeled cells enumerated by flow cytometry. The percentage of specific killing was calculated as follows: 1 $-[(\% \text{CFSE}^{lo} \text{control}/\% \text{CFSE}^{hi} \text{control})/(\% \text{CFSE}^{lo} \text{treated}/\% \text{CFSE}^{hi} \text{treated})] \times$ 100. For adoptive cell transfer experiments, RMA-S or RMA-bearing mice received one infusion of CD8$^+$ T cells ($0.25 \times 10^6$) 2 days after tumor implantation. CD8$^+$ T cells infused in RMA-S-bearing mice were isolated from the MC38-bearing mice as described below. CD8$^+$ T cells infused in RMA-bearing mice were isolated from the RMA-bearing mice after two doses of Nucl-siRNA conjugates. Cells from tumor-draining lymph nodes were isolated and restimulated in vitro during 48 h with IL-2 (20 IU/ml) in the presence of irradiated RMA-S-B7 (1:10, APC:target ratio) and autologous splenocytes (1:1, splenocytes:target ratio). CD8$^+$ T cells were purified using a MACS-negative selection column (Miltenyi Biotec).

MC38 colon adenocarcinoma model. Protocol was used as described in ref. [21]. Briefly, 7–9-week-old female C57BL/6 mice were inoculated with $1 \times 10^5$ MC38 tumor cells s.c. and treatment was initiated 6–7 days after inoculation (palpable tumors with volume of ~25–75 mm³). Adjuvant (50 μg anti-CD40 Ab clone FJK45 plus 100 μg poly(I:C) (InvivoGen)) in PBS or adjuvant with 50 μg Reps1, Adpgk and Dpagt1 peptides each, were administered i.p. Treatment schedule for Nucl-TAP siRNA was the same as used for the subcutaneously implanted models. Peptides were purchased from GenScript and sequences were as follows Reps1: GRVLELFRAAQLANDVVLQIMELCGATR; Adpgk: GIPVHLELASMTNMELMS SIVHQQVFPT; Dpagt1: EAGQSLVISASIIVFNLLELEGDYR. For the generation of TAP-deficient specific CD8$^+$ T cells, MC38-bearing mice that have received two doses of Nucl-siRNA conjugates as described for the 4T1 model were euthanized 2 days after the second dose. Cells from tumor-draining lymph nodes were isolated and restimulated in vitro as described for RMA-bearing mice. For in vitro T cell reactivity assays, CD8$^+$ T cells were cultured with DC 2.4 cells pulsed with THR4 or FAP (FAPLRPLPTL peptides (1 μg/ml)[24]. DC 2.4 cells pulsed with an irrelevant peptide for H-2Db (Ad10) or for Qa-1b (FL9, FYAEATPML, Nagarajan et al. (Nat Immunol 2012)) were used as negative control of TAP-deficient specific CD8$^+$ T cell activation. Murine IFNγ production after 20 h stimulation was measured by ELISA from Thermo Fisher Scientific.

Mice implanted with the transplantable tumors were sacrificed when tumor diameter exceeded 12 mm, mice exhibited signs of morbidity, or ulcerated tumors. Experiments were terminated when two or more mice were sacrificed in the "untreated" group, except for combination experiments in 4T1 tumor model and experiments using MC38 tumor model.

Orthotopic model for pancreatic cancer[19]. 7–9-week-old female C57BL/6J mice were anesthetized, laparotomy was performed, short-term cultured KPC-derived cells ($2 \times 10^3$) mixed with pancreatic stellate cells ($1.8 \times 10^4$) were injected into the tail of the pancreas. The pancreas was then carefully returned to the peritoneal cavity, the abdomen was closed with a 4–0 vicryl suture, and the skin was stapled. Ten days following tumor implantation, Nucl-siRNAs were administered i.p at 0.5 nmol per dose (0.875 mg/ml) twice per week for 3 weeks. Minnelide was given at 0.21 mg/kg/day by i.p. injection daily. The animals were sacrificed when animals in controls groups were showing signs of morbidity. Tumor weight and the presence of liver, peritoneal, and mesenteric metastases were determined.

BRAF/PTEN melanoma model: $Braf^{CA}Pten^{loxP}Tyr::CreER^{T2}$ mice on the C57BL/6 background were crossed with $Pten^{lox/lox}$ mice to generate $Braf/Pten$-TG mice[52]. For localized melanoma inductions, 5 weeks old male and female mice were topically treated with 4-Hydroxytamoxifen (4-HT) (Sigma). Two microliters of 5 mg/ml HT in DMSO was administered to the right ear on 2 consecutive days. Starting 2 weeks following induction, mice were treated (i.p.) twice a week for 4 weeks with 0.5 nmol Nucl-TAP (0.875 mg/ml), PD-1 antibody (100 µg), and Flt3 ligand (10 µg). Note that treatment regimen was more intense compared to other models because of the challenging nature of this model. Mice were euthanized when pigmented lesions covered >90% of the ear.

**Cell lines**. A20, 4T1, GL261, Chinese hamster ovary (CHO) and NIH3T3 cells were purchased from ATCC. MC38 were purchased from Kerafast. Immortalized dendritic cell line DC2.4 mouse dendritic cell line was purchased from Millipore Sigma. D2SC1 murine dendritic cell line was kindly provided by Dr. Paula Ricciardi-Castagnoli[53], RMA, RMA-S (TAP2-deficient), and RMA-S.B7.1 (RMA-S transfected with mouse CD80 gene) cells were described before[12,13]. The isolation and culture of cells derived from primary KPC tumors and pancreatic stellate was described before[19]. A panel of HLA-A*02:01-positive melanomas and renal cell carcinomas and their TAP1 knock-out variants generated by CRISPR/CAS9 technology was previously described[28]. The generation and culture of mouse CD8+ T cell clone LnB5, with specificity for the TRH4-derived peptide MCLRMTAVM in the context of H-2Db and the isolation and expansion of human CD8+ T cell clone 1A8 specific for the LRPAP1-derived peptide in the context of HLA-A2 were previously described[11–13,28]. The TCR of the human T cells was cloned into a pMP71-TCR-flex backbone and retrovirally transduced into pre-stimulated CD8+ T cells of a healthy donor according to previously described procedures[54]. D2SC1 dendritic cell line was transduced with lentiviral vector (PTIG-U6tetOshRNA) encoding murine TAP2 or control shRNAs expressed from a tet-regulated U6 promoter[55]. shRNA expression was upregulated in vitro by adding doxycycline to the culture medium. Doxycycline-induced murine TAP2 shRNA expression in cultured D2SC1 cells resulted in down-regulation of the murine TAP2 mRNA.

**Culture conditions**. Cell lines were cultured in RPMI-1640 medium (A20, 4T1, RMA, and DC2.4 cells), Dulbecco's modified Eagle's medium (MC38, GL261, and NIH3T3), F12 (CHO cells) or Iscove's modified Dulbecco's medium (human tumor cells and T cell activation assays) from Gibco, supplemented with 8–10% heat-inactivated FCS, 100 U/ml penicillin, and 100 µg/ml streptomycin. RMA variants were additionally supplemented with 1 mM sodium pyruvate, 0.05 mM β-mercaptoethanol, and 1× minimal essential medium (MEM) non-essential amino acids. All cell lines and assay cultures were maintained at 37 °C and 5% $CO_2$. All cells were tested regularly for mycoplasma contamination.

**Design and characterization of Nucleolin aptamer-siRNAs conjugates**. Nucleolin or control aptamer[17], extended at the 3' end with the following sequence (termed linker): 5' GUACAUUCUAGAUAGCC, were purchased from Trilink Biotechnologies. Complementary linker sequences extending from the sense strand of murine TAP2 (5'GCUGCACACGGUUCAGAAT), murine ERAAP (5' GCUAUUACAUUGUGCAUTA), human TAP1 (5' CAGGAUGAGUUACUU-GAAA) or control (Ctrl) (5' UAAAGAACCAUGGCUAACC) siRNAs were ordered from IDT and contained 2' Omethyl modified pyrimidines with the last two bases being deoxynucleotides. Antisense siRNA sequences, ordered from IDT, were as follows: murine TAP2 (5' AUUCUGAACCGUGUGCAGCmUmU), murine ERAAP (5' UAAUGCACAAUGUAAAUAGCmUmU), human TAP1 (5' UUUCAAGUAACUCAUCCUGmUmU) and Ctrl (5' GGUUAGCCAUG-GUUCUUUAmUmU) whereby 'm' indicated the presence of an O'-methyl modified ribonucleotide. For mRNA targeting siRNAs characterization, candidate sequences were predicted using HPC Dispatcher (City of Hope, Biomedical Informatics Core, Duarte, CA, USA), siRNA scales (Department of Human Genetics, University of Utah, Salt Lake City, UT, USA), and siDESIGN (Dharmacon, Thermo Fisher Scientific) software. Overlapping predictions featuring a low melting temperature ($T_m$) were selected and screened for specific activity as nucleolin aptamer conjugates using ΨCheck assay in transiently transfected CHO cells[56]. Aptamers were annealed to duplex siRNAs in PBS at 37 °C for 10 min in a block heater and allowed to cool to room temperature.

**siRNA knockdown in vitro**. Cells were plated in triplicates onto 24-well plates ($2.5–5 \times 10^4$ cells) for 18 h. After complete adhesion, cells were incubated with

0.5 µM of Nucl-siRNA conjugates two times every 8 h. For apoptosis determination, 24 h after the last treatment, an Annexin V/7-amino-actinomycin D (7AAD) double staining was carried out according to manufacturer's protocol (BD Biosciences). For quantitative PCR, 24 h after the last treatment, RNA was isolated using an RNeasy kit (QIAGEN). RNA was quantified using an Agilent 2100 Bioanalyzer (Agilent Technologies). cDNA synthesis was performed using the High Capacity cDNA Reverse Transcription kit (Applied Biosystems). cDNA equivalents of 25–50 ng of mRNA were used per reaction in a TaqMan qPCR assay using the Step One qPCR machine (Applied Biosystems), with primer sets corresponding to the gene of interest or housekeeping products. For flow cytometer analysis, cells were harvested 24, 48, 72 or 96 h, after the last treatment and stained as described below. For in vitro T cell reactivity assays, cells were harvested 48 or 72 h after the last treatment, and cultured with activated Lnb5 T cells, CD8+ T cells from MC38-bearing mice treated with Nucl-siRNAs, or TAP-deficient specific human CD8+ T cells. RMA cells pulsed with THR4 peptide (1 µg/ml) or RMA-S were used as positive control of LnB5 T cell activation. TAP-deficient human tumor cells were used as positive control of TAP-deficient specific human CD8+ T cell activation. Murine or human IFNγ production after 20 h stimulation was measured by ELISA fromThermo Fisher Scientific or R&D systems, respectively. Cytotoxic activity was determined in 4 h in vitro lactate dehydrogenase assay (Thermo Fisher Scientific). Percentage of specific lysis was calculated as: ([experimental release−effector cell release−spontaneous release]/[maximum release−spontaneous release]) × 100. Cell viability after 24, 48, and 72 h culture with Nucl-siRNA conjugates was measured by MTT assay. Percentage of viable cells calculation: (AbsΔ 540–620 nm of treated cells/AbsΔ 540–620 nm of untreated cells) × 100.

**Flow cytometry and cell sorting**. Multicolor flow cytometry staining was performed using the following antibodies and reagents: CD45-FITC (30-F11), CD3-BV785 (17A2), CD19-APC/Cy7 (6D5), CD8a-BV510 (53-6.7), CD8-APC (53-6.7), CD4-AF700 (RM4-5), CD335-BV605 (29A1.4), Tim3-APC (RMT 3.23), F4/80-APC (BM8), CD11c-PE/Cy7 (N418), I-A/I-E-BV785 (M5/114.15.2), Ly-6C-PerCP/Cy5.5 (HK1.4), CD31-BV421 (390), H-2Dd-FITC (34-2-12), H-2Kd-PerCP/Cy5.5 (SF1-1.1), H-2Kb-FITC (AF6-88.5), H-2-PE (M1/42), Ly-6G-BV605 (1A8), HLA-ABC-PE (W6/32), CD24-APC (30-F1), CD64-PE (X54-5/7.1), CD49b-PerCP/Cy5.5 (DX5), KLRG1-APC (2F1/KLRG1), CD44-PerCP/Cy5.5 (IM7), CD69-BV421 (H1.2F3), and PE Streptavidin from BioLegend. CD3-APC/Cy7 (17A2), Ly-6G-APC/Cy7 (1A8), and CD11b-redFluor710 (M1/70) were obtained from Tonbo Biosciences. CD279-BV421 (J43), CD19-PF-594 (1D3), CD140a-PE/Cy7 (APA5), CD103-BV421 (M290), CD137-BV421 (1HA2), and biotin-Qa-1[b] (6A8.6F10.1A6) were obtained from BD Biosciences. CD25-PE/Cy7 (PC61.5), CD137-PE (17B5), and Foxp3-PE (FJK-16s) were obtained from eBiosciences. TRH4 tetramer (TRH4 Tetramer/PE H-2Db) was produced by the central protein facility of the Leiden University Medical Center. Tumor-infiltrating lymphocytes (TILs) or tumor cells were analyzed as previously described; briefly, cells were isolated by dissecting tumor tissue into small pieces followed by incubation in 1 mg/ml of collagenase and 100 µg/ml of DNase in complete RPMI-1640 medium prior to dissociation using the gentleMACS Dissociator (Miltenyi Biotec). Cell suspension was passed through a 70 µm nylon strainer to obtain a single-cell population. Cells were washed twice with FACS buffer (PBS pH 7.2, supplemented with 0.5% BSA, 2 mM EDTA, and 0.09% azide) and stained as described below. Cells were incubated with Fc-blocking antibodies (purified anti-mouse CD16/32, clone 93, Biolegend) for 10 min at 4 °C, then incubated with antibodies for 30 min at 4 °C. A20-GFP tumors for MHC class I analysis or cell sorting were processed using the same protocol. For tetramer experiments, cells were stained with the tetramer for 10 min at 22 °C, followed by staining with CD8 and CD19 antibodies. Cell suspensions were incubated with fixable viability dye eFluor780 (eBioscience) for 15 min at 4 °C. After a washing step, cells were either analyzed directly or fixed with 4% PFA (BD) solution for 30 min and stored in a 1% PFA solution until analysis. For intracellular staining of Foxp3, cells were fixed and permeabilized with Foxp3 Staining Buffer Kit (eBioscience) according to manufacturer's instructions. Cells were analyzed using a Beckman Coulter's CytoFLEX flow cytometer or LSR II cytometer (Becton Dickinson) and data were analyzed using FlowJo 10 software (TreeStar). Cell sorting was performed using Beckman Coulter MoFlo Astrios EQ and cells were collected in complete RPMI-1640 medium for RNA isolation and quantitative PCR as described previously. Cytokine concentration in serum was analyzed using a CytoFLEX flow cytometer and LEGENDplex software (BioLegend).

**Tumor homing and biodistribution of [32]P-labeled aptamers**. [32]P-labeled Nucleolin or control aptamers were generated by annealing to a complimentary oligonucleotide that was synthesized in the presence of αP[31] as previously described with sequence: 5'-AUUCUGAACCGUGUGCAGCACCACCGCUGCACACGGUUCAGAAUACACA**CGAGGCTATCTAGAATGTAC**, whereby the bold region indicates complementarity to the nucleolin aptamer linker sequence. The non-complementary sequence corresponds to the TAP siRNA where both strands are held together via a short loop. A total of $5 \times 10^5$ cpm of material was injected i.p. into mice-bearing 4T1 tumors that were ~150 mm³. Mice were sacrificed 24 h post injection. Naïve mice were used as controls. Tumor, lung, heart, liver, spleen, kidney, and TDLN were excised and washed extensively with PBS.

Next, tumors and organs were weighed and placed in 500 ml of scintillation fluid, and counted with a scintillation counter.

**Toxicology studies**. Two in vivo experiments were performed. First, Balb/c mice treated with six doses of Nucl-siRNAs every 3 days were sacrificed 2 days after the last dose. Blood was collected from mice after three and six doses. Second, T1-tumor-bearing mice treated with Nucl-siRNAs, PD-1 Ab, Flt3 ligand, or CTLA4 Ab as described in immunotherapy studies were sacrificed 2 days after the last dose and blood was collected. Complete blood count (CBC) analysis was performed using HemaVet analyzer (Drew Scientific). Liver enzymes AST and ALT were quantified in serum by using a colorimetric Aspartate Aminotransferase Activity Assay Kit (Sigma) or Alanine Aminotransferase Activity Assay Kit (Cayman Chemical), respectively, according to the manufacturer's protocol. Cytokines were quantified in serum by LEGENDplex Mouse Inflammation panel (13-plex; Bio-Legend) according to the manufacturer's recommendations. Lung, liver, spleen, kidney, liver, and TDLN weights were determined. Lung, liver, and small intestine were fixed in 10% formalin overnight at room temperature and embedded in paraffin. Slides were stained with eosin (Across organics) for 5 min, followed by 10 min of hematoxylin staining (Vector) incubated with 0.1% sodium bicarbonate as a bluing agent, rinsed in $H_2O$. Slides were dehydrated in increasing concentrations of alcohol (70%, 90%, 100%; 3 min each), briefly washed in xylene, and coverslipped using mounting medium (Richard-Allan Scientific). CD4/CD8 T cells ratio and activation was analyzed in spleens by flow cytometry as described before.

**Quantification and statistical analysis**. When variables studied were normally distributed, statistical analysis of multiple comparisons was performed using one-way ANOVA with Tuckey or Dunnet posttest. To compare the mean differences between groups that have been split into two independent variables (treatment/number of doses), analyses were performed using two-way ANOVA. Non-parametrical methods were applied for not normally distributed variables. For these statistical analyses, multiple comparisons was performed using Kruskall–Wallis with Dunn posttest, and comparisons between just two groups were performed using Mann–Whitney $U$-test. Significance of overall survival was determined via Kaplan–Meier analysis with log-rank (Mantel–Cox) analysis. All statistical analyses were performed with Graphpad Prism 6 and 7 (GraphPad). Error bars show standard error of the mean (SEM), and $P < 0.05$ was considered statistically significant. *Indicates $P < 0.05$, **$P < 0.01$, ***$P < 0.005$, and ****$P < 0.001$ unless otherwise indicated.

**Reporting summary**. Further information on research design is available in the Nature Research Reporting Summary linked to this article.

## Data availability

Data supporting the findings of this study are available within the paper and its supplementary information files. In the event of inadvertent omission, the additional data will be available upon request from the corresponding author.

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

## Acknowledgements

We thank Paula Bates and E. Merit Reyes-Reyes for guidance in using the Nucleolin aptamer, Oliver Umland, and Patricia Guevara for assistance with flow cytometry and sorting, Gabe Gaidosh for help with confocal imaging. We acknowledge support from the Miami Center for AIDS Research (CFAR) at the University of Miami Miller School of Medicine which is funded by a grant (P30AI073961) from the National Institutes of Health (NIH), and the Flow Cytometry Shared Resource of the Sylvester Comprehensive Cancer Center at the University of Miami, Miller School of Medicine. This work was supported by the Dodson foundation and the Sylvester Comprehensive Cancer Center, Medical School, University of Miami.

## Author contributions

E.G. is principal investigator, conceptualized the study and experimental strategy and co-wrote the manuscript. G.G. planned the experiments and co-wrote the manuscript. B.S. helped experimental planning performed immune therapy experiments in transplantable models. A.R. and F.E. performed the immune analysis. A.L. performed the experiment in BRAF/PTEN model. A.B. and A.L developed and characterized the murine Nucl-TAP siRNAs. T.G. and A.L. developed and characterized the human TAP siRNAs. S.M. and V.D. performed the experiments in the PDA model. E.D., K.M. and T.v.H. generated mouse and human CD8 T cell clones and planned the experiments.

## Additional information

**Competing interests:** E.G. is a founder and equity holder of Sebastian Biopharma that holds intellectual rights for the neoantigen induction approach described in this study. The remaining authors declare no competing interests.

