## [Peer Review File · Nature Communications]

Reviewers' Comments:

Reviewer #1:

Remarks to the Author:

This is an interesting manuscript showing that in vivo treatment with a Tap-reduction causing treatment that is selective for a variety of transplantable mouse tumors substantially retards the growth of these tumors, but does not cure them in most cases. This could be an intriguing approach to cancer immunotherapy, but a number of questions remain:

1) In the introduction the authors are critical of the difficulties associated with exploiting mutation-derived neo-epitopes, as if the spontaneous response to these neo-epitopes is more optimal than they suggest. However in the discussion the same is brought forward by the authors for TAP-negative tumors. In both instances immunotherapy directed against the neo-epitopes or the TEIPP epitopes can be used to improve the immunotherapeutic effects. Thus the authors should be less dismissive of the therapy directed against neo-epitopes. In fact they are encouraged to explore whether simultaneous therapy directed against neo-epitopes and TEIPP epitopes is synergistic. Have they generated such data?

2) What is the duration of MHC class I downregulation caused by the anti-TAP therapy in vivo?

3) To which extent induced the TAP downregulation epitope spreading towards other epitopes

4) In Fig. 4A anti-CD4+ anti-CD8 treatment has the same effect as anti-Asialo antibody treatment, both completely abolishing the anti-Tumor effect. This should be explained

5) The authors must explore the possibility that the in vivo anti-tumor effects are caused by detection of TEIPP epitopes on non-treated tumor cells, so that the T cells against these epitopes, once induced by the treatment, crossreact to minute quantities of these epitopes on non-treated tumor cells or tumor cells in which the effects of the aptamers have worn off. If this were the case, the value of the potential clinical application rises. As we know, MHC class I processing in many tumor cells is suboptimal, even in the absence of overt TAP downregulation. It is also known that many fewer MHC-peptide complexes are needed for target cell recognition than for initiation of the response and the TEIPP T cells could therefore still be tumor-selective

Reviewer #2:

Remarks to the Author:

Garrido et al. build on the pioneering work of the van Hall laboratory in demonstrating the unique and useful features of TAP-independent generation of MHC class I restricted immunogenic peptides. This is a very cool study in which a nucleolin-binding aptamer is used to deliver TAP-specific shRNA to tumor cells. The effect is remarkably specific in vitro, and even in vivo. The authors clearly show that this leads in both human and mouse cells to the presentation of TAP-independent peptides and the immune mediated destruction of tumors in mice, based on both CD8+ T cell and NK effector activity. All experiments are well controlled, and the thorough and thoughtful Discussion is a joy to read. All in all, a terrific paper that will be of wide interest with important potential for tumor immunotherapy.

Reviewer #3:

None

REVIEWERS COMMENTS & AUTHORS RESPONSES:

Reviewer #1 (Remarks to the Author):

Legends to revised MS:

New text (significantly modified from original)	highlighted in blue;
Text removed from original version	strike-through.
Original text brought to the attention	highlighted in yellow.

This is an interesting manuscript showing that in vivo treatment with a Tap-reduction causing treatment that is selective for a variety of transplantable mouse tumors substantially retards the growth of these tumors, but does not cure them in most cases. This could be an intriguing approach to cancer immunotherapy, but a number of questions remain:

Reviewer has noted that the approach we are describing in this study “*retards the growth of these tumors, but does not cure them in most cases*”. We wish to point out that it was not our expectation that this, or for that matter any other immune potentiating strategy, will be curative as monotherapy. Drawing from the correlation between responsiveness to checkpoint blockade and (neo)antigenicity of the patient tumors, this study describes a way to enhance tumor antigenicity in situ and hence enhance the susceptibility of otherwise poorly antigenic tumors to immune potentiating strategies, in all likelihood not limited to checkpoint blockade therapy.

Responses to specific comments

1) In the introduction the authors are critical of the difficulties associated with exploiting mutation-derived neo-epitopes, as if the spontaneous response to these neo-epitopes is more optimal than they suggest. However in the discussion the same is brought forward by the authors for TAP-negative tumors. In both instances immunotherapy directed against the neo-epitopes or the TEIPP epitopes can be used to improve the immunotherapeutic effects. Thus the authors should be less dismissive of the therapy directed against neo-epitopes. In fact they are encouraged to explore whether simultaneous therapy directed against neo-epitopes and TEIPP epitopes is synergistic. Have they generated such data?

We concur, and as per the reviewer’s suggestion we toned down the discussion on challenges associated with targeting tumor resident neoantigens in terms of their identification and potential for inducing immune dysfunction (the corresponding sections were removed from the ABSTRACT, INTRODUCTION, and DISCUSSION). We have refocused the rationale of our approach to address one key and well recognized limitation of resident neoantigens – their paucity in many tumors (e.g. Fig. 1 in reference 3), and that consequently, as well documented, many patients are less likely to benefit from checkpoint blockade therapy, and conceivably from other forms of immune therapy. Our approach – as discussed in the manuscript – overcomes this limitation and could, therefore, broaden the responsiveness of patients’ immune therapy to include also the many patients that do not express or express to few endogenous neoantigens.

We have not tested a combination of vaccination against neoepitopes and induction of TAP downregulation induced antigens. We are planning to do that as part of a study to evaluate the immune dominance of the TAP downregulation induced antigens versus “classical” neoantigens. We hypothesize that the former are dominant (indicated by the side-by-side comparison, Fig. 3G, compare untreated to Nucl-TAP treated groups), and that combination will not significantly enhance tumor inhibition. Remains to be seen...

2) What is the duration of MHC class I downregulation caused by the anti-TAP therapy in vivo?

As suggested by the reviewer, we evaluated the duration of MHC class I downregulation and found that it persisted for about 4 days. We also confirmed these observations by measuring the downregulation of TAP mRNA using qRT-PCR (Figure; modified text on p.6). Thus, systemic treatment with Nucl-TAP siRNA, especially if repeated several times, should provide sufficient time for activating cognate T cells, and lead to the biological effects in terms of immune responses and tumor inhibition described in this study. We consolidated these results in the new Figure 2 of the revised manuscript.

Figure: GFP+A20 tumor bearing mice were treated with Nucl-TAP siRNA and at the indicated time tumors were excised. TAP mRNA levels were determined by qRT-PCR on GFP+ sorted cells and MHC class Ia expression was determined by flow cytometry on gating on GFP+ cells. (For additional details see new Figure 2).

3) To which extent induced the TAP downregulation epitope spreading towards other epitopes

We thank the reviewer of raising this point. Epitope spreading is thought to be indicative of an increasingly potent vaccine effect. We, therefore, designed and carried out an experiment in two models that indeed suggests that tumor targeted downregulation of TAP induces epitope spreading. CD8⁺ T cells were isolated from RMA tumor bearing mice that were treated with Nucl-TAP or Nucl-Ctrl siRNA and transferred to RMA tumor bearing mice that were or were not treated with Nucl-TAP siRNA. As shown in the Figure below, the adoptively transferred CD8⁺ T cells from Nucl-TAP, but not Nucl-Ctrl, siRNA treated mice inhibited tumor growth in tumor bearing mice that were not treated with Nucl-TAP siRNA, but was enhanced upon Nucl-TAP siRNA treatment. Thus, inducing new antigens in tumor cells by systemic treatment with Nucl-TAP siRNA elicited also T cell responses that recognized the parental tumor cells that did not present the induced antigens. This experiment shown below in panel A was included in the revised MS (text page 8, new Fig. 4K).

To further evaluate the existence of epitope spread we used a second experimental model whereby CD8⁺ T cells isolated from 67NR breast carcinoma tumor bearing mice treated with either Nucl-TAP or Nucl-Ctrl siRNA were transferred to recipient 67NR tumor bearing mice (not treated with Nucl-TAP or Nucl-Ctrl siRNAs). Consistent with epitope spread, we saw (panel B) that T cells from Nucl-TAP, but not Nucl-Ctrl, siRNA treated mice inhibited tumor growth. As a positive control, treatment of 67NR tumor bearing mice with Nucl-TAP, but not Nucl-Ctrl, siRNA inhibited tumor growth. This experiment was also included in the revised MS (Page 8, new Fig. 4L)

Figure: *TAP downregulation mediated induction of non-TAP deficient specific CD8+ T antitumor immunity.* Adoptive transfer of CD8+ T cells from RMA (A) or 67NR (B) bearing mice treated with Nucl-TAP inhibits the growth of these tumors in the absence of Nucl-TAP treatment. Black arrow, T cells infusion; red arrows, Nucl-siRNAs treatment (6-9 mice/group).

4) In Fig. 4A anti-CD4+ anti-CD8 treatment has the same effect as anti-Asialo antibody treatment, both completely abolishing the anti-Tumor effect. This should be explained

While the role of the adaptive CD8 and CD4 T cell response was to be expected, we elaborated on the observed importance of NK cells discussing why NK cells could play a role in TAP downregulation mediated immunity(text/page 8 & Discussion page 15), and followed up with additional experiments to support the role and mechanism of action of NK cells (Fig. 4B-D).

5) The authors must explore the possibility that the in vivo anti-tumor effects are caused by detection of TEIPP epitopes on non-treated tumor cells, so that the T cells against these epitopes, once induced by the treatment, crossreact to minute quantities of these epitopes on non-treated tumor cells or tumor cells in which the effects of the aptamers have worn off. If this were the case, the value of the potential clinical application rises. As we know, MHC class I processing in many tumor cells is suboptimal, even in the absence of overt TAP downregulation. IT is also known that many fewer MHC-peptide complexes are needed for target cell recognition than for initiation of the response and the TEIPP T cells could therefore still be tumor-selective

We saw no clear evidence to suggest that the TAP TEIPP T cells induced by downregulation of TAP in tumor cells recognize TAP-sufficient normal or tumor cells presenting minute amounts of TAP TEIPP to activated T cells. The evidence against it, arguably not conclusive, is two-fold:

(1) TAP TEIPP specific murine and human CD8+ T cell clones recognize TAP downregulated, but not parental, targets; our data from this manuscript whereby TAP was downregulated with Nucl-TAP siRNA (Fig. 1C & D; Fig. 6), and previous published studies by one of us (TvH) using genetically deficient murine or human T cells (for example, reference 13 and 29).

That said, a slightly higher level of IFN γ secretion from untreated or Nucl-Ctrl treated cells compared to medium (Fig. 1C) could represent a low level of TAP TEIPP presentation by normal tumor cells. Whether a low level of TAP TEIPP presentation by normal cell contributes to better protective immunity (in mice treated with Nucl-TAP siRNA) would be not easy to determine. .

(2). We see no toxicity in mice treated with tumor (nucleolin)-targeted TAP siRNA under condition that we observe tumor inhibition (Fig 5E & F), suggesting, indirectly, that the T cells generated against TAP TEIPP do not recognize normal cells that might have expressed minute amounts of TAP TEIPP – at least not to the extent that it could be clinically important.

A scenario that the reviewer’s point would apply – and augment the value of this approach – stems from observations that in some instances progressing tumors sporadically downregulate TAP in, albeit a proportions, of tumor cells within a given lesion. However, that would be very hard to model in mice.

Reviewer #2 (Remarks to the Author):

Garrido et al. build on the pioneering work of the van Hall laboratory in demonstrating the unique and useful features of TAP-independent generation of MHC class I restricted immunogenic peptides. This is a very cool study in which a nucleolin-binding aptamer is used to deliver TAP-specific shRNA to tumor cells. The effect is remarkably specific in vitro, and even in vivo. The authors clearly show that this leads in both human and mouse cells to the presentation of TAP-independent peptides and the immune mediated destruction of tumors in mice, based on both CD8+ T cell and NK effector activity. All experiments are well controlled, and the thorough and thoughtful Discussion is a joy to read. All in all, a terrific paper that will be of wide interest with important potential for tumor immunotherapy.

Reviewers' Comments:

Reviewer #1:

Remarks to the Author:

The authors have adequately responded to the points raised and substantially and satisfactorily improved the manuscript